# Dynamic Multi-Mode Mie Model for Gain-Assisted Metal Nano-Spheres

**DOI:** 10.3390/ma16051911

**Published:** 2023-02-25

**Authors:** Nicole Recalde, Daniel Bustamante, Melissa Infusino, Alessandro Veltri

**Affiliations:** 1Colegio de Ciencias e Ingenieria, Universidad San Francisco de Quito, Quito 170901, Ecuador; 2Department of Physics, Boston University, Boston, MA 02215, USA

**Keywords:** localized surface plasmons, gain assisted metal nanoparticles, Mie theory

## Abstract

Coupling externally pumped gain materials with plasmonic spherical particles, even in the simplest case of a single spherical nanoparticle in a uniform gain medium, generates an incredibly rich variety of electrodynamic phenomena. The appropriate theoretical description of these systems is dictated by the quantity of the included gain and the size of the nano-particle. On the one hand, when the gain level is below the threshold separating the absorption and the emission regime, a steady-state approach is a rather adequate depiction, yet a time dynamic approach becomes fundamental when this threshold is exceeded. On the other hand, while a quasi-static approximation can be used to model nanoparticles when they are much smaller than the exciting wavelength, a more complete scattering theory is necessary to discuss larger nanoparticles. In this paper, we describe a novel method including a time-dynamical approach to the Mie scattering theory, which is able to account for all the most enticing aspects of the problem without any limitation in the particle’s size. Ultimately, although the presented approach does not fully describe the emission regime yet, it does allow us to predict the transient states preceding emission and represents an essential step forward in the direction of a model able to adequately describe the full electromagnetic phenomenology of these systems.

## 1. Introduction

The idea of coupling metal nanoparticles (mNP) with gain elements [1,2,3,4,5,6] is almost coeval to the very dawn of the stunning scientific interest born around the beginning of the century about the possibility to support localized surface plasmons (LSP) in these nanostructures [7,8,9]. During the last twenty years, the study of hybrid nano-resonators including metal and gain elements (mgNP), rather than remain bound to the role of an exotic sub-topic, constituted an imposingly growing parallel field [10,11,12,13,14,15,16,17,18,19]. Undoubtedly, this was partially due on account of how, while one of the most promising applications of LSPs was to allow the development of visible metamaterials, this very possibility was prevented by the large amount of losses the most viable metals show at these frequencies. However, while the pursuit to bring spectacular features provided by metamaterials such as cloaking [20,21,22] and super-lenses [23,24] in the visible range surely spotlighted mgNPs, the interest in these structures was not at all limited to this aspect. This research field was in fact, in the very same years, spawning novel theoretical ideas such as the SPASER [25,26,27], which rapidly evolved from a controversial proposal to an experimental reality [6,28,29] and is now finding its way to real life applications [30,31,32]. In this dynamic landscape, we focused our interest on spherically symmetric mgNPs, discussing theoretical properties [33,34,35] and proposing possible new applications [36,37] of both the core–shell (particles where a metal core is embedded in a gain enriched dielectric shell) and the nano-shell (a metallic shell covering a gain enriched core) geometries. Meanwhile, we also realized that even the simplest case of a single metal nanoparticle in a uniform gain medium [33] still conceals a rich and yet untapped physical phenomenology: in fact, in a previous work [34], we discussed how the widely used steady state approach correctly describes the system only when a quantity of gain below a threshold is involved. When higher gain levels are introduced in the system, the structure begins to emit and the steady state approach fails. In the same work, we introduced a more complex time-dynamical approach able to handle the emissive regime. However, that preliminary analysis was limited to the quasi-static limit (where the size of the nanoparticle is much smaller than the exciting wavelength).

In this paper, we present a theoretical description of the same system, this time transcending any particle size limit by means of a time-dynamic Mie theory able to capture the most fascinating aspects of the problem. This new and original approach allows us, in principle, to follow the evolution in time of any of the infinite multipolar resonances in the scattered electromagnetic field of a mgNP. Moreover, by including a technique derived from the Optical Bloch equations, this model also allows us to describe the interplay between the chosen gain element and the time evolution of each multipolar resonance. In what follows, we will validate the results of this model by comparing them with the ones obtained with the quasi-static approach presented in [34]. We will then present a novel characterization that was out of reach without the extension presented here and use it to identify a relation between the steady state behavior of a mode with the possibility to turn it into an electromagnetic emitter. Finally, as a working example of the wide range of novel characterizations this method can allow, we will demonstrate how it is possible (by choosing the right gain emission center-line) to inject energy in a weak/higher mode, effectively “sculpting” the shape of the resulting scattered field.

## 2. Constitutive System of Equations

In a previous work [34], we discussed how the time dependence of the polarization in a metal $P_{m}$ and in a gain medium $P_{h}$ can be calculated as: (1)$$\begin{array}{cc} \; & P_{h}\left(t\right)=\varepsilon_{0}\chi_{b}E_{h}+\Pi_{h}\left(t\right), \end{array}$$(2)$$\begin{array}{cc} \; & P_{m}\left(t\right)=\varepsilon_{0}\chi_{\infty }E_{m}+\Pi_{m}\left(t\right); \end{array}$$
where $\varepsilon_{0}$ is the vacuum permittivity, $E_{h}$ and $E_{m}$ are the electric fields, respectively, in the gain medium and in the metal, $\chi_{b}$ is the susceptibility of the dielectric in which the gain elements are dissolved and $\chi_{\infty }$ is the susceptibility due to the metal ions. The time dependence of the polarizations comes from the dynamic quantities $\Pi_{h}$ and $\Pi_{m}$, which are determined by the following system of equations: (3)$$\begin{array}{cc} \; & \frac{d\Pi_{h}}{dt}-\left[i\left(\omega -\omega_{21}\right)-\frac{1}{\tau_{2}}\right]\Pi_{h}=-\frac{iGN}{\tilde{N}\tau_{2}}E_{h}, \end{array}$$(4)$$\begin{array}{cc} \; & \frac{dN}{dt}+\frac{N-\tilde{N}}{\tau_{1}}=-\frac{i}{2}\left(\Pi_{h}\cdot E_{h}^{*}-\Pi_{h}^{*}\cdot E_{h}\right), \end{array}$$(5)$$\begin{array}{cc} \; & \frac{d\Pi_{m}}{dt}-\frac{\omega (\omega +2i\gamma )}{2(\gamma -i\omega )}\Pi_{m}=\frac{1}{2(\gamma -i\omega )}E_{m}, \end{array}$$
where $\omega_{21}$ is the center-line for the gain emission, $\tau_{1}$ is the effective relaxation time, $\tau_{2}$ is the time constant associated with relaxation processes of the gain element (which is related to their emission bandwidth Δ through $\Delta =2/\tau_{2}$), while $\tilde{N}$ is the asymptotic population inversion (i.e., the fraction of the gain element population that would be pumped in the excited state if no saturation mechanism was in play). In system (Equation 3)–(Equation 5) we also used the following normalization for field and polarization:$$E\to \sqrt{\frac{\varepsilon_{0}}{n\hbar \omega_{\mathrm{pl}}}}E\;\;P\to \frac{P}{\sqrt{\varepsilon_{0}n\hbar \omega_{\mathrm{pl}}}}$$
where *n* is the gain elements molecular density, *ℏ* the reduced Planck constant and $\omega_{\mathrm{pl}}$ the plasma frequency of the metal. Additionally, frequencies are normalized to $\omega_{\mathrm{pl}}$, time is normalized to $1/\omega_{\mathrm{pl}}$, and length to the radius *a* of the nanoparticle. Finally, the normalization for the transition dipole moment was chosen as:$$\mu \to \frac{\mu }{a^{3}\sqrt{6\varepsilon_{0}n\hbar \omega_{\mathrm{pl}}}}.$$

The quantity *G* in Equation (Equation 3) represents the amount of gain present in the system, and it can be calculated as:(6)$$G=\mu^{2}\tau_{2}n^{2}\tilde{N}$$

### 2.1. Rotational Fields and How Too Much Gain Breaks the Quasi-Static Limit

It is well established [38] that the quasi-static limit (in which the only resonating mode of a spherical nanoparticle is dipolar) works only for nanoparticles much smaller than the exciting wavelength; however, in the model we presented in [34], we have shown that this approximation also breaks for these small particles when too much gain is present in the system. Specifically: when the gain level of the system is higher than the emission threshold $G_{\mathrm{th}}$, a multipolar cascade is always produced, and thus a pure dipolar emission is never allowed in this geometry. This effect can be functionally discussed in terms of the rotationality of the involved polarizations. Specifically, in [34] we defined the equivalent of a “potential” for the the dynamical part of the polarizations:(7)$$\Pi_{m,h}=-\nabla \psi_{m,h}\;;$$
this is accurate only if these vector fields are irrotational:(8)$$\nabla \times \Pi_{m,h}=0.$$

We can evaluate the solidity of this assumption by applying the curl to Equations (Equation 3) and (Equation 5) and taking into account that
$$\nabla \times \left(NE_{h}\right)=N\nabla \times E_{h}-E\times \nabla N\;\;\mathrm{and}\;\mathrm{that}\;\;\nabla \times E_{m,h}=0;$$
which will produce the system of equations regulating the time dependence of the curl of the polarizations in the gain medium ($\nabla \times \Pi_{h}$) and in the metal ($\nabla \times \Pi_{m}$):(9)$$\begin{array}{cc} \; & \frac{d}{dt}\nabla \times \Pi_{h}-\left[i\left(\omega -\omega_{21}\right)-\frac{1}{\tau_{2}}\right]\nabla \times \Pi_{h}=\frac{iG}{\tilde{N}\tau_{2}}E_{h}\times \nabla N \end{array}$$(10)$$\begin{array}{cc} \; & \frac{d}{dt}\nabla \times \Pi_{m}-\frac{\omega (\omega +2i\gamma )}{2(\gamma -i\omega )}\nabla \times \Pi_{m}=0. \end{array}$$Equation (Equation 10) is a first order homogeneous differential equation in $\nabla \times \Pi_{m}$ meaning that, if this quantity is zero at t=0, it will stay zero forever (and also that, if for some exotic reason it was something other than zero at t=0, it will decay to zero after a short transient). While this clearly ensures the irrotationality of $\Pi_{m}$ and consequently allows for the definition of the potential $\psi_{m}$ (as in Equation (Equation 8)), the same is much less straightforward in the case of Equation (Equation 9). Here, one can use the same line of reasoning only if ∇N is zero, corresponding to a uniform population inversion N(r)=N. This means that $\Pi_{h}$ is irrotational, and the definition of $\psi_{h}$ is allowed *as long as N does not depend on the spatial coordinates*: when ∇N≠0, the driving term
$$\frac{iG}{\tilde{N}\tau_{2}}E_{h}\times \nabla N$$
of Equation (Equation 9) becomes non-negligible, the equation turns non-homogeneous, and $\Pi_{h}$ is forced out of irrotationality. A realistic initial condition for *N* is for it to be zero everywhere, then (assuming a spatially uniform pump) in a time comparable with $\tau_{1}$, the second term of Equation (Equation 4) drives it everywhere to $\tilde{N}$. A closer look at Equation (Equation 4) shows that the spatial dependency of *N* is produced through the left hand term
$$-\frac{i}{2}\left(\Pi_{h}\cdot E_{h}^{*}-\Pi_{h}^{*}\cdot E_{h}\right),$$
which can be shown to be proportional to the intensity of the electric field in the gain region. The extensive characterization we have carried out of the solutions of system (Equation 3)–(Equation 5) reveals that this term can in fact be neglected as long as the gain level *G* of the system is lower than the threshold needed to produce emissions. Physically, this is understandable because that term is a saturation term for *N*, which becomes non-negligible only when the plasmonic field absorbs energy from the gain medium faster than the pump can replenish it. Since, in the case of a single nanoparticle in an infinite gain medium, the field is dipolar in the gain region, the depletion of *N* is faster around the nanoparticle’s poles than it is everywhere else. This phenomenon, known as Spatial Hole Burning (SHB), produces a spatial dependency in the population inversion and thus *breaks the irrotationality* of the dynamical part of the polarization in the gain medium. This means that, as soon as the system enters the emissive regime, the pseudo-potentials model we presented in [34] surrenders and cannot be used to properly describe the system anymore.

### 2.2. Polarization Projections on Vector Spherical Harmonics

The mandatory next step in the pursuit of a full multipolar depiction of gain assisted metal nanoparticles in spherical symmetry is to extend the model in order to include particles of any size. The natural way to achieve this is to use the projection of the electric fields on the Vector Spherical Harmonics $M_{e1n}$, $M_{o1n}$, $N_{e1n}$, and $N_{o1n}$ used in Mie Theory. In this description [38], the incident field $E_{\mathrm{inc}}$ is modeled as a plane wave whose projection reads as:(11)$$E_{\mathrm{inc}}\left(r,t\right)=\sum_{n=1}^{\infty }E_{n}\left[M_{o1n}^{\left(1\right)}\left(r\right)-iN_{e1n}^{\left(1\right)}\left(r\right)\right]$$
while the internal $E_{\mathrm{in}}$ and the scattered $E_{\mathrm{sca}}$ fields can be written, respectively, as: (12)$$\begin{array}{cc} E_{\mathrm{in}}\left(r,t\right) & =\sum_{n=1}^{\infty }E_{n}\left[c_{n}\left(t\right)M_{o1n}^{\left(1\right)}\left(r\right)-id_{n}\left(t\right)N_{e1n}^{\left(1\right)}\left(r\right)\right], \end{array}$$(13)$$\begin{array}{cc} E_{\mathrm{sca}}\left(r,t\right) & =\sum_{n=1}^{\infty }E_{n}\left[ia_{n}\left(t\right)N_{e1n}^{\left(3\right)}\left(r\right)-b_{n}\left(t\right)M_{o1n}^{\left(3\right)}\left(r\right)\right]. \end{array}$$In all of the Equations (Equation 12) and (Equation 13), we have that
$$E_{n}=E_{0}i^{n}\frac{2n+1}{n(n+1)},$$
where $E_{0}$ is the amplitude of the incident field, the upper indices (1) and (3) also refer to the Bessel function used to describe the radial dependency [38].

From expressions (Equation 11)–(Equation 13) and using the superposition principle, one can easily obtain the following forms for the field inside the metal particle $E_{m}$ and for the external field in the gain medium $E_{h}$:
(14)$$\begin{array}{cc} E_{m}\left(r,t\right) & =\sum_{n=1}^{\infty }E_{n}\left[c_{n}\left(t\right)M_{o1n}^{\left(1\right)}\left(r\right)-id_{n}\left(t\right)N_{e1n}^{\left(1\right)}\left(r\right)\right], \end{array}$$
(15)$$\begin{array}{cc} E_{h}\left(r,t\right) & =\sum_{n=1}^{\infty }E_{n}\left[M_{o1n}^{\left(1\right)}\left(r\right)-iN_{e1n}^{\left(1\right)}\left(r\right)+ia_{n}\left(t\right)N_{e1n}^{\left(3\right)}\left(r\right)-b_{n}\left(t\right)M_{o1n}^{\left(3\right)}\left(r\right)\right]. \end{array}$$

The core hypothesis of our model is to assume a similar shape for the dynamical part of the polarizations: (16)$$\begin{array}{cc} \Pi_{m}\left(r,t\right) & =\sum_{n=1}^{\infty }\left[\kappa_{n}\left(t\right)M_{o1n}^{\left(1\right)}\left(r\right)-i\delta_{n}\left(t\right)N_{e1n}^{\left(1\right)}\left(r\right)\right], \end{array}$$(17)$$\begin{array}{cc} \Pi_{h}\left(r,t\right) & =\sum_{n=1}^{\infty }\left[\zeta_{n}\left(t\right)M_{o1n}^{\left(1\right)}\left(r\right)-i\eta_{n}\left(t\right)N_{e1n}^{\left(1\right)}\left(r\right)+i\alpha_{n}\left(t\right)N_{e1n}^{\left(3\right)}\left(r\right)-\beta_{n}\left(t\right)M_{o1n}^{\left(3\right)}\left(r\right)\right]. \end{array}$$It might be worth mentioning that this gauge essentially substitutes the irrotationality assumption, which was necessary to define the polarization potentials. Substituting (Equation 14)–(Equation 17) into system (Equation 3)–(Equation 5), one obtains a system of equations for the time evolution of the polarization coefficients: the equations for $\zeta_{n}$ and $\eta_{n}$, accounting for the continuity with the external field, are independent and can be solved analytically. The remaining four equations read as: (18)$$\begin{array}{cc} \; & \frac{d\alpha_{n}}{dt}-\left[i\left(\omega -\omega_{21}\right)-\frac{1}{\tau_{2}}\right]\alpha_{n}=-\frac{iGN}{\tilde{N}\tau_{2}}E_{n}a_{n}, \end{array}$$(19)$$\begin{array}{cc} \; & \frac{d\beta_{n}}{dt}-\left[i\left(\omega -\omega_{21}\right)-\frac{1}{\tau_{2}}\right]\beta_{n}=-\frac{iGN}{\tilde{N}\tau_{2}}E_{n}b_{n}, \end{array}$$(20)$$\begin{array}{cc} \; & \frac{d\kappa_{n}}{dt}-\frac{\omega (\omega +2i\gamma )}{2(\gamma -i\omega )}\kappa_{n}=\frac{1}{\left(\gamma -i\omega \right)}c_{n}, \end{array}$$(21)$$\begin{array}{cc} \; & \frac{d\delta_{n}}{dt}-\frac{\omega (\omega +2i\gamma )}{2(\gamma -i\omega )}\delta_{n}=\frac{1}{\left(\gamma -i\omega \right)}d_{n}. \end{array}$$Radial and tangential continuity, together with the definition of the displacement vectors in the two regions
(22)$$\begin{array}{cc} D_{m} & =\varepsilon_{\infty }E_{m}+\Pi_{m} \end{array}$$
(23)$$\begin{array}{cc} D_{h} & =\varepsilon_{b}E_{h}+\Pi_{h} \end{array}$$
allow writing $a_{n}$, $b_{n}$, $c_{n}$ and $d_{n}$ as functions of the coefficients for the polarizations. Additionally, Mie theory needs a working definition for the indexes of refraction of the involved media in order to calculate the wave numbers in the two regions; for this, we used relations (Equation 22) and (Equation 23) and calculate the time dependent dielectric permittivities $\varepsilon_{m}\left(t\right)$ and $\varepsilon_{h}\left(t\right)$ as: (24)$$\begin{array}{cc} \varepsilon_{m} & =\varepsilon_{\infty }+\frac{\Pi_{m}\cdot E_{m}}{|E_{m}{|}^{2}}, \end{array}$$(25)$$\begin{array}{cc} \varepsilon_{h} & =\varepsilon_{b}+\frac{\Pi_{h}\cdot E_{h}}{|E_{h}{|}^{2}}; \end{array}$$
from which we can define the adimensional radial variables:
(26)$$\begin{array}{cc} \rho_{m} & =rk_{m}=\frac{r\omega \sqrt{\varepsilon_{m}}}{c} \end{array}$$
(27)$$\begin{array}{cc} \rho_{h} & =rk_{h}=\frac{r\omega \sqrt{\varepsilon_{h}}}{c}, \end{array}$$
which must be used to ensure the radial continuity at each time step. This tangled coupling mixes up the equations of system (Equation 18)–(Equation 21) in a way that makes an analytical approach somewhat cumbersome, the numerical solution of the same system, however, is rather straightforward and will be presented in the next section.

## 3. Numerical Solutions

### 3.1. Quasi-Static Limit and Beyond

We first validated the model by comparing the Mie polarizability
(28)$$\alpha_{Mie}=6\pi \frac{ia_{1}}{k^{3}},$$
where $a_{1}$ is calculated through the numerical solutions of system (Equation 18)–(Equation 21) with the dynamical quasi-static polarizability $\alpha_{QS}$ defined in the model [34] in the case of nanoparticles much smaller than the wavelength. These two quantities were calculated for different levels of gain in the case of a silver nanoparticle with the unrealistic radius of 1 nm dissolved in a water solution of gain elements whose central emission frequency was superposed with the plasmon resonance frequency.

As one can see in Figure 1a,b, when no gain is added to the system, the time evolution frequency-by-frequency of $\alpha_{Mie}$ and $\alpha_{QS}$ overlaps perfectly (Figure 1a), and both time dependencies converge to the corresponding value on the steady state spectrum (Figure 1b). In Figure 1c,d, we show the very same flawless correspondence when some gain, below the emission threshold, is added to the system. Furthermore, when the gain in the system exceeds the emission threshold as in Figure 1e,f, even if the correspondence with the spectrum is lost for both $\alpha_{Mie}$ and $\alpha_{QS}$, their time dependencies overlap perfectly. It might be worth mentioning here that, consistently with our previous results [34,39], both models confirm that a negative value of the imaginary part of the steady state polarizability (Figure 1f) corresponds to an emissive regime in the time dependent description (Figure 1e).

When used in a range where the quasi-static approximation is known to break, the model presented in this work produces different results which, as one can see in Figure 2 in the case of a nanoparticle of radius 60 nm, converge to the corresponding value of the steady state spectrum both when no gain is added to the system (Figure 2a,b) and with gain below the emission threshold (Figure 2c,d). We do not report here on the behavior over threshold because it would not add to the validity of the new model.

### 3.2. Time Dependence of Mie Coefficients

The main advantage of the new model is that it transcends the quasi-static limit and therefore it is not limited to the description of the time-dependence of the dipolar mode; this enables it to potentially characterize the time evolution of every coefficient of the Mie expansions (Equation 14)–(Equation 15), thus allowing for the dynamical depiction of all the optical properties of nanoparticles of any size. In order to present some of the capabilities of this new feature, we will discuss in this section the time evolution of the first two coefficients of the Mie expansion for the scattered field (namely $a_{1}$ and $a_{2}$) for a 60 nm radius silver nanosphere in water. As mentioned in the previous section, a particle of this size is beyond the quasi-static limit; however, as one can see in Figure 3, the amplitude of the resonance related to the quadrupolar mode ($a_{2}$ in Figure 3d) is still more than two orders of magnitude smaller than the one of the dipolar mode ($a_{2}$ in Figure 3b). Figure 3 shows the time evolution of the first (Figure 3a) and the second (Figure 3c) coefficient, each one calculated for a frequency close to their respective resonance frequency. As expected, both of their steady states correspond to those calculated using the classical Mie theory one can find in the literature [38].

When gain is added to such a system, its behavior strongly depends on the chosen emission central frequency $\omega_{21}$. In Figure 4a–d, we centered the emission of the gain elements on the resonance frequency of the dipolar mode. One can appreciate that even in this case, the time dependence of the real and the imaginary parts of $a_{1}$ (Figure 4a) and $a_{2}$ (Figure 4c) converge to the corresponding values in the steady state spectra (Figure 4b,d). However, by comparing Figure 4b with Figure 3b, and Figure 4d with Figure 3d, one can easily notice that, while the energy injected through the gain elements produces a threefold increase in the intensity of the dipolar resonance, the quadrupolar one barely changes at all.

On the other hand, as we show in Figure 4e–h, if the gain element emission frequency $\omega_{21}$ is positioned in correspondence with the quadrupolar mode resonance, the coefficient $a_{2}$ is the one that is most affected by the injection of gain into the system. As one can see in Figure 4e,f, while an effect of the gain medium on the dipolar mode is still present, it is definitely less intense than the one obtained when the gain was positioned in correspondence with the dipolar resonance (see Figure 4a,b): by comparing Figure 3b with Figure 4f, one can notice a slight shift of the dipolar resonance in the direction of the gain emission frequency and some small enhancement of the dipolar resonance. The most important change here happens on the higher mode: by comparing Figure 4h with Figure 3d, one can identify a sixfold enhancement of the quadrupolar resonance intensity, demonstrating that a negligible mode can be turned into a relevant one, by wisely choosing the gain emission centerline. It is worth noting here that the quadrupolar mode seems to be more sensitive than the dipolar one to the gain of the surrounding media (i.e., a sixfold increase vs. a threefold one), this could be due to the non-linear nature of the amplification process or a spectral integral effect related to the quadrupolar mode being thinner. A wider and more specific characterization is needed to properly address the nature of this difference: a following publication is planned in this direction. It is fundamental to stress that this effect of gain-driven, higher-mode enhancement happens with sub-emissive quantities of gain ($G<G_{\mathrm{th}}$), and although it shares the ability to activate higher modes (that would be otherwise negligible in small particles) with the gain-driven mode-cascade [34] we discussed in Section 2.1, they are two very distinct effects that must not be confused with one another.

The strong relation between the chosen emission center-line of the gain element and the resonant frequency of the mode also means that, “in principle”, one should be able to drive the emission of a higher mode by means of a super-emissive quantity of gain ($G>G_{\mathrm{th}}$) without major distortions on the electromagnetic field of the lower modes. When this happens, the model presented in this paper is not able to adequately describe the resulting electromagnetic behavior because, as discussed in Section 2.1, emission corresponds to a spatial and temporal dependency in the population inversion *N*, which is not yet included in this approach. However, we can use the same reasonable considerations we presented in [34] for the dipolar mode to infer that a mode cascade will also occur when higher modes are driven to emission. This means that, when the emission begins, we cannot ensure the original mode will be preserved or in any way predict the final shape of the electromagnetic field. Nonetheless, one can still use the presented model to identify a mechanism similar to the one we have singled out for the dipolar mode in the quasi-static limit [33,34,39] (i.e., when the imaginary part of polarizability goes negative, an emissive regime emerges). Here, when a generic mode is involved, emission appears as soon as the *real part of the Mie scattering coefficient* relative to that mode becomes negative. One can use this relation between the real part of the scattering coefficient and the emission regime to evaluate the threshold gain $G_{\mathrm{th}}$ needed to drive each mode into emission. As an example, in the case of a silver nanoparticle of the radius 60 nm in water, when the gain center-line is chosen to correspond to the resonance frequency of the quadrupolar mode, the gain needed to drive that mode to emission can be estimated as $G_{\mathrm{th}}=0.03495$; this means that, if a larger quantity of gain is used (e.g., G=0.04, as in the calculation presented in Figure 5), the real part of the quadrupolar coefficient $a_{1}$ will be negative in a nonzero spectral region (Figure 5b) and the corresponding time evolution of the same coefficient will diverge (Figure 5a). A wider characterization of the mode-dependent emissive regimes of spherical metal nanoparticles is currently under preparation.

### 3.3. Sub Emissive Gain-Driven Field “Sculpturing”

In order to better describe the relation between the injection of gain at a certain frequency and its effect on the shape of the scattered field, in this section we will present the streamlines of the scattered electric field $E_{\mathrm{sca}}$ calculated using Equation (Equation 13) and keeping in mind that $a_{n}∼0$ for n>2 and $b_{n}∼0$ for n>1. All the calculations presented in this section are performed for a 60 nm radius silver nanoparticle hosted in water.

In Figure 6a, we can see that, in the absence of gain and when plotted in the same scale as $a_{1}$, the spectrum of $a_{2}$ is basically imperceptible, meaning that, even at the very resonance frequency of the quadrupolar mode, the dipolar coefficient $a_{1}$ still dominates. This is reflected in the shape of the scattered field calculated for the same frequency and presented in Figure 6b; here, one can notice that, while the quadrupolar resonance can in fact produce some disturbances on the field shape, these are limited to a region very close to the nanoparticle and the field keeps everywhere else an almost perfect dipolar structure.

We already discussed at the end of the previous section that a negligible scattering mode can be made relevant by nourishing it with energy provided by a gain element emitting at the resonance frequency of the relative Mie coefficient. The effectiveness of this amplification depends on the quantity of gain *G* injected into the system. In order to quantify this effect, we show in Figure 7 the square amplitudes of $a_{1}$ and $a_{2}$ as a function of the gain quantity *G* (plotted up to $G_{\mathrm{th}}$) for the resonance frequency of $a_{2}$, which was also chosen as the emission center-line $\omega_{21}$ of the gain element. Here, one can clearly see that, as *G* increases, $|a_{1}{|}^{2}$ remains almost constant (i.e., in the logarithmic scale used for this plot), while $|a_{2}{|}^{2}$ *climbs up six orders of magnitude*. Consistently with what we discussed in the previous section, an opposite outcome would have been obtained if the gain emission center-line $\omega_{21}$ was chosen closer to the dipolar resonance frequency.

**Figure 6 materials-16-01911-f006:**
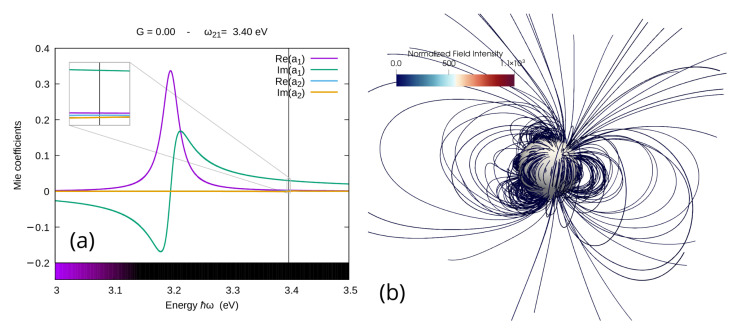
(**a**) Spectra for the real and imaginary part of the first two coefficients of the Mie expansion of the scattered field for a silver nanoparticle of radius 60 nm in water when no gain is added to the system, the inset in the upper left corner is a zoom around the resonance frequency of the second coefficient; (**b**) streamlines of the scattered field calculated for the quadrupolar central frequency ℏω=3.4 eV, and the colorbar range here is the same as Figure 8b.

Using a simple bisection method, it is fairly easy to find that $|a_{2}{|}^{2}$ surpasses $|a_{1}{|}^{2}$ at $G_{s}=0.03334$, which is still lower than the emission threshold $G_{\mathrm{th}}=0.03495$ calculated in the previous section for the same system. This means that we have a working *G* range in which the energy injected at the quadrupolar mode is enough to make it dominant, but still not enough to drive it to emission. In Figure 8, we present the effect of a gain quantity G=0.0345 chosen in this range: one can see in Figure 8a that, around its resonance frequency, the coefficient $a_{2}$ became visible and larger than the dipolar coefficient $a_{1}$. This dominance of the second Mie coefficient is reflected on the scattered field shape around the nanoparticle which, as one can appreciate in Figure 8b, now renders a purer quadrupolar shape.

**Figure 8 materials-16-01911-f008:**
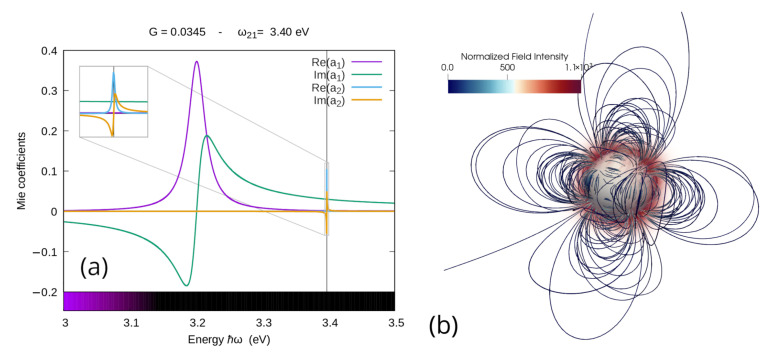
(**a**) Spectra for the real and imaginary part of the first two coefficients of the Mie expansion of the scattered field for a silver nanoparticle of radius 60 nm in water enriched with a gain medium below the emission threshold G=0.03 with emission bandwidth Δ=0.15 eV. Here, the emission central frequency $\hbar \omega_{21}=3.4$ eV was chosen to correspond to the resonance frequency of the quadrupolar mode, the inset in the upper left corner is a zoom around the resonance frequency of the second coefficient; (**b**): streamlines of the scattered field calculated for the quadrupole central frequency, and the colorbar range here is the same as Figure 6b.

It is worth noting here that, while we can efficiently use our model to predict that this effect manifests for *G* higher than $G_{s}$, and that a proper quadrupolar field shape is kept until $G_{\mathrm{th}}$, it does not allow us to discuss the range $G>G_{\mathrm{th}}$. We know in fact that, when the system enters the emissive regime, the resulting mode cascade, discussed in Section 2.1 and Section 3.2, could potentially destroy the quadrupolar field shape. A model including a proper description of the temporal and spatial dependency of the population inversion *N* must be developed in order to predict the field dynamics in this regime.

## 4. Discussion

In a previous work [34], we have shown how a SHB phenomenon can produce a mode cascade when the system is excited with a gain level beyond the one needed to produce emissions. In this work, we discussed how this mode cascade occurs when the model presented in our previous work breaks down because, as soon as the SHB kicks in, the dynamical parts of the polarization becomes irrotational, thus preventing the definition of the pseudo-potentials we used in the previous model.

While the model presented here is not yet capable of describing this SHB induced mode cascade, it is definitely a necessary step in the right direction because it allows us to discuss multi-modal plasmonic fields and, as a consequence, it already expands the possibilities of the previous one by allowing us to account for nanoparticles of any size. While proving the validity of this novel approach and presenting the additional capabilities it brings to the table, we also spotlighted how, by introducing gain at the proper frequency range, one can enhance a higher mode using a sub-emissive quantity of gain. This was identified another way, in addition to the SHB mode cascade, allowing us to change the shape of the scattered fields.

There are obvious parallels between these two phenomena; however, they are completely unrelated and produced by distinct physical effects also: (a) the mode cascade is not related to a specific frequency and can happen to any mode when it is driven to emission; (b) the sub-emissive high mode enhancement needs the gain to be added at the specific frequency range of the specific mode one wants to enhance; (c) in order to enhance a higher mode by means of a sub-emissive gain quantity, the mode (however small) “has to be there from the beginning”, whereas the mode cascade can happen even in the smallest particle possible, where before the cascade the only active mode is the dipolar one.

Using the novel capabilities of this new model, we also identified a relation between the real part of the steady state Mie scattering coefficient turning negative and an emissive regime manifesting in the time domain. This can be considered an extension of the relation between polarizability and emission we discussed in previous papers [22,34,39]. Once a mode begins to emit, however, the model presented here is not able to adequately describe the electromagnetic behavior of our system because, in this case as well, a mode cascade has to be expected.

For these reasons, a further exploration of the phenomena arising from the interplay between plasmonic resonances and active gain elements requires an additional extension of this model. Namely: one has to include the ability to manage the non-homogeneous population inversion that arises in the emissive regimes. This will single-handedly allow us to depict complex electromagnetic phenomenology induced by the mode cascade.

## 5. Conclusions

The interplay between the plasmonic resonance of metal nanoparticles and the ability to inject energy in a specific frequency range provided by active gain materials is a virtual land of undisclosed phenomena of which we have but touched the shores. Even in the apparently simplest scenario of a metal sphere in an infinite gain medium, the parameter space to explore is much vaster than one could have imagined: one of the key features in this interaction is related to the capability of gain to activate higher mode resonances even in particles much smaller than their resonance wavelength, for which it was believed the only sustainable mode was dipolar.

This work represents a clear step forward in understanding, modeling and predicting the behavior of a Mie mgNP in both the sub-emissive and emissive regimes. We discussed in depth the limit of applicability of the model in its present form, especially in relation to the mode cascade arising due to the SHB when the gain level exceeds the emission threshold. A new phenomenon, occurring in the sub-emissive regime of gain coupled mMP, has also been predicted: when gain is injected at the desired frequencies, higher-order and less intense modes can be selectively enhanced, allowing for a tailored spatial shaping of the electromagnetic field around the particle. This phenomenon can be uniquely observed in the Mie regime, where the presence of a multipolar behavior gives multiple possibilities for the spatial shaping of the field, even with at a sub-emissive amount of gain.

## Figures and Tables

**Figure 1 materials-16-01911-f001:**
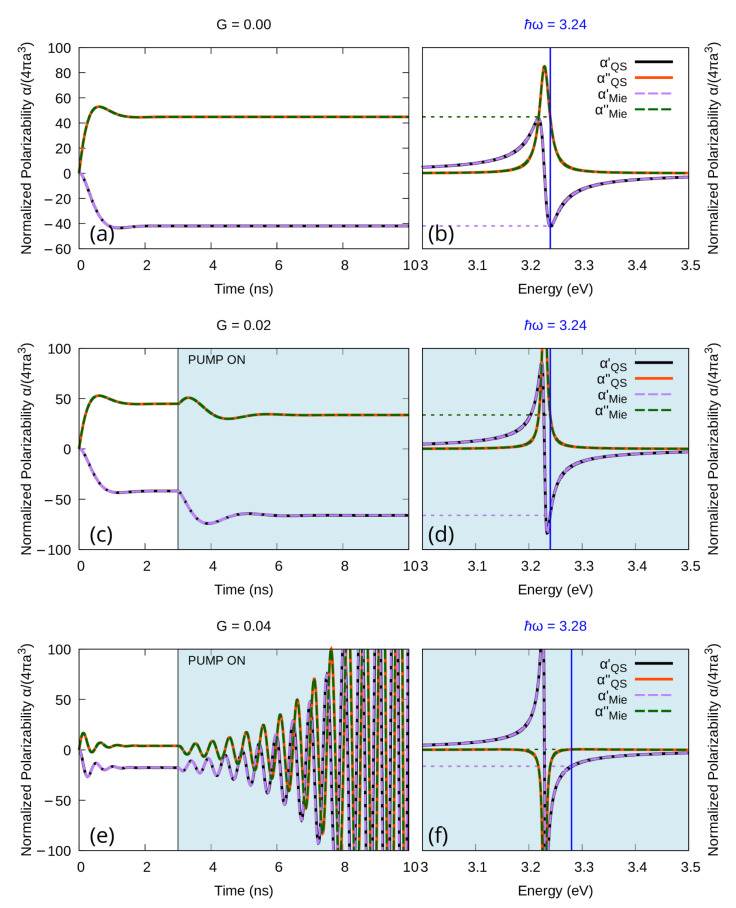
Real and imaginary part of the Polarizability of a silver nanoparticle of radius 1 nm in water enriched with a gain medium with emission central frequency $\hbar \omega_{21}=3.23$ eV and emission width Δ=0.15 eV. (**a**,**b**): No gain is added to the system. (**c**,**d**): Some gain is included, but the system remains sub-emissive ($G=0.5\cdot G_{\mathrm{th}}$). (**e**,**f**): Enough gain is included to drive the system to the emissive regime ($G=1.1\cdot G_{\mathrm{th}}$). (**a**,**c**,**e**): Time dependence of the polarizability for a single frequency; (**b**,**d**,**f**): corresponding steady state spectrum.

**Figure 2 materials-16-01911-f002:**
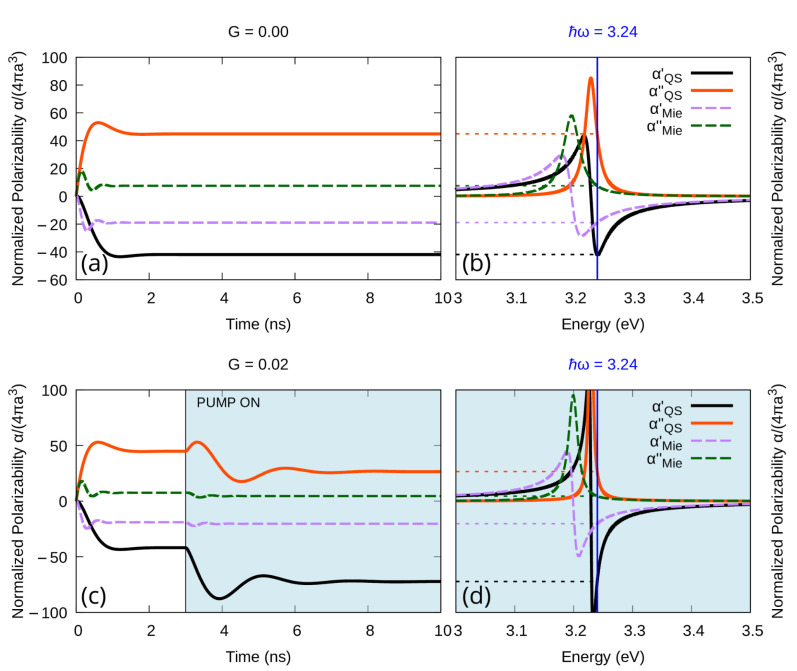
Real and imaginary part of the Polarizability of a silver nanoparticle of radius 60 nm in water enriched with a gain medium with emission central frequency $\hbar \omega_{21}=3.23$ eV and emission width Δ=0.15 eV. (**a**,**b**): No gain is added to the system. (**c**,**d**): Some gain is included, but the system remains sub-emissive ($G=0.5\cdot G_{\mathrm{th}}$). (**a**,**c**): Time dependence of the polarizability for a single frequency; (**b**,**d**): corresponding steady state spectrum.

**Figure 3 materials-16-01911-f003:**
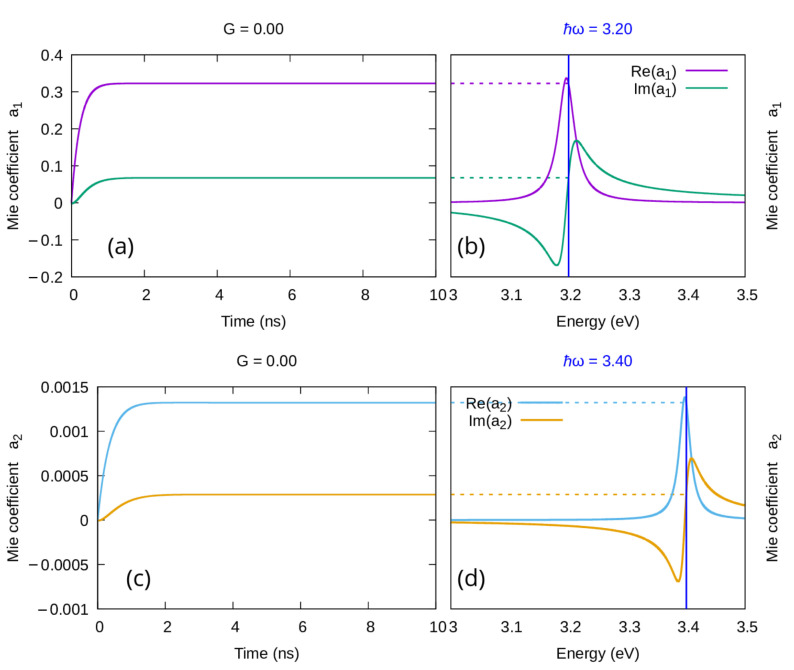
Real and imaginary parts of the first two coefficients of the Mie expansion of the scattered field for a silver nanoparticle of radius 60 nm in water, when no gain is included in the system. (**a**,**b**): coefficient $a_{1}$ dipolar mode; (**c**,**d**): coefficient $a_{2}$ quadrupolar mode; (**a**,**c**): time dependencies for a single frequency; (**b**,**d**): corresponding steady state spectrum.

**Figure 4 materials-16-01911-f004:**
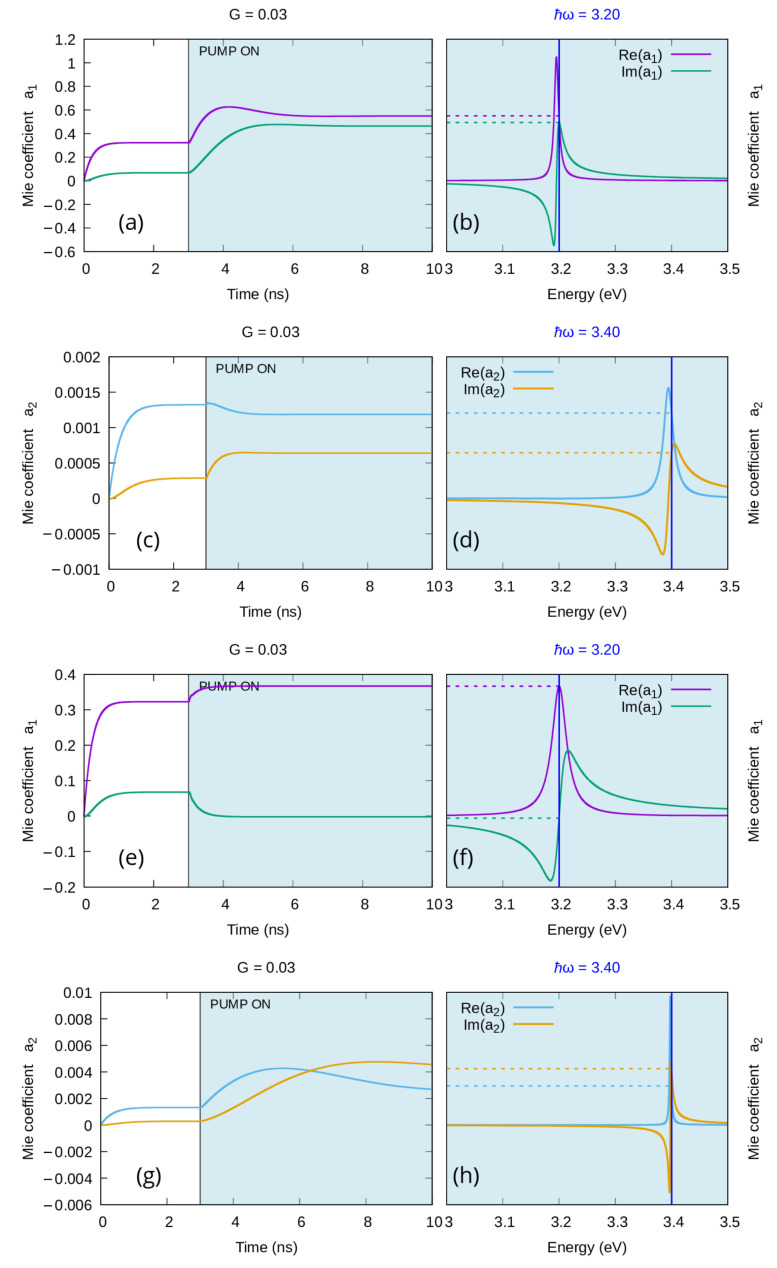
Real and imaginary part of the first two coefficients of the Mie expansion of the scattered field for a silver nanoparticle of radius 60 nm in water enriched with a gain medium below the emission threshold G=0.03 with emission bandwidth Δ=0.15 eV. (**a**–**d**): the emission central frequency $\hbar \omega_{21}=3.19$ eV was chosen to correspond to the resonance frequency of the dipolar mode. (**e**–**h**): the emission central frequency $\hbar \omega_{21}=3.4$ eV was chosen to correspond to the resonance frequency of the quadrupolar mode. (**a**,**b**,**e**,**f**): coefficient $a_{1}$ dipolar mode; (**c**,**d**,**g**,**h**): coefficient $a_{2}$ quadrupolar mode; (**a**,**c**,**e**,**g**): time dependencies for a single frequency; (**b**,**d**,**f**,**h**): corresponding steady state spectrum.

**Figure 5 materials-16-01911-f005:**
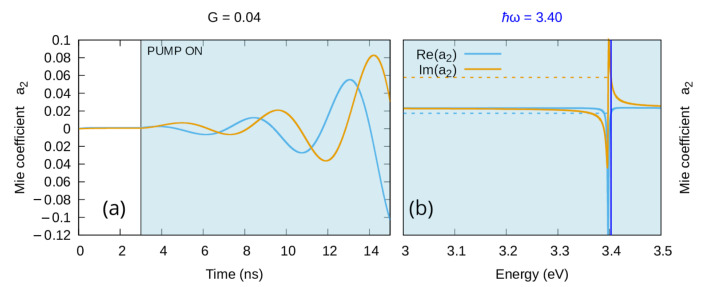
Real and imaginary part quadrupolar coefficient of the Mie expansion of the scattered field for a silver nanoparticle of radius 60 nm in water enriched with a gain medium above the emission threshold G=0.04 with emission bandwidth Δ=0.15 eV. Here, the emission central frequency $\hbar \omega_{21}=3.4$ eV was chosen to correspond to the resonance frequency of the quadrupolar mode. (**a**): Time dependencies for a single frequency; (**b**): corresponding steady state spectrum.

**Figure 7 materials-16-01911-f007:**
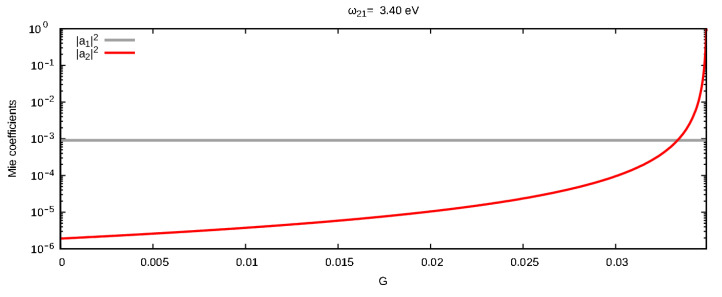
Behavior of the square modulus of the first two coefficients of the Mie expansion for the scattered field at the quadrupolar mode frequency; calculated for a silver nanoparticle of radius 60 nm in water as a function of the gain quantity *G* present in the system. Here, the emission central frequency was chosen to correspond to the resonance frequency of the quadrupolar mode.

## Data Availability

Not applicable.

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
