# Peer review of "Dynamic Multi-Mode Mie Model for Gain-Assisted Metal Nano-Spheres"

_materials, 2023, doi:10.3390/ma16051911_

Round 1

Reviewer 1 Report

This manuscript investigates the phenomenon of spherical particles creating an “incredibly rich electrodynamic phenomenology even in the simplest case of a single spherical nanoparticle in a uniform gain medium”. The appropriate theoretical description of these systems is dictated by the quantity of the included gain and by the size of the nano-particle. I think the methodology and the approach was very interesting. I do recommend this paper and think for theoretical approaches in this field, this could be a very good paper. One minor issue: Any reason conclusion does not exist in the paper?

Author Response

We appreciate the commentary given by Reviewer 1 on the relevance and approach of our work. As they suggest, a conclusion was added to the paper.

Reviewer 2 Report

The authors presented a thorough theoretical analysis of electrodynamic properties of a metal spherical particle embedded in a gain-enriched medium using Mie scattering theory and accounting for time-dependencies of polarizations. This manuscript is an extension to the previous work of the authors [https://doi.org/10.1038/srep33018], where they used quasi-static (dipole) approximation applicable only for small particles. Contrary, the present analysis is valid for an arbitrary size of the particle and accounts for multimode expansion of the emission. Authors demonstrated and discussed in details validity of their approach. Also, they showed some novel results this approach brings, e.g., a sub emissive quantity of gain in the surrounding medium leads to enhancement of a quadrupolar mode (and, presumably, higher ones). The authors admit that the model still does not properly describe mode cascade induced by spatial hole burning phenomenon that occurs for higher quantities of gain. However, the present work is a step in the right direction.

The subject under discussion is actual. This is evident from the references, almost all of which are from 2000s-2010s and some are from 2020s. The manuscript is scientifically well-written, and the conclusions are mostly reasonable. Though, the content is a little too methodical, in my view. Also, graphical representation of the data could be improved. Overall, with some corrections the manuscript seems as a publishable piece of work, which can be interested for a broad audience in fields of plasmonics, nanophotonics and light-matter interactions. Below I list several comments and corrections, mostly minor, which, in my opinion, are required to make the manuscript better:

1)      I believe, a reader, who is less familiar with Mie Theory than the authors, would appreciate some comments about physical meaning of the expansions represented as Eqs. (11-12), particularly about expansion coefficients and which terms stand for an incident wave. The fact that the incident wave is a plane one should be mentioned. Also, upper indices (1) and (3) of vector spherical harmonics should be explained. Please, make the abovementioned corrections and additions.

Please, correct a typo in Eq. (11): there is a duplication E(r,t)(r,t)=…

2)      All the figures presented demonstrate time-dependencies and spectra of real and imaginary parts of polarizability/Mie coefficients a1,2. Are both real and imaginary parts are necessary for the representation? Would only the modulus of these values suffice? This would greatly relieve the graphics (twice less lines). Though, I leave this at the discretion of the authors.

3)      As I understood, Figure 1 is dedicated to show that for a small particle the model developed provides the same result as the quasi-static approximation. Though, the red and black lines, which stand for the quasi-static polarization, are not seen in the graphs. I understand, they just exactly coincide with the purple and blue lines. To avoid confusions, I suggest the authors to improve this representation, e.g. change the style of some lines to dashed or dotted. This way, all the lines will be seen above one another. 

4)      As I understood, the only purpose of Figure 3 is to show that for a 10-nm-particle a quadrupolar term is much less than a dipolar one. This is a pretty evident statement. Does it require a whole separate figure with 4 parts, two graphs in each? I suggest to shorten this section.

5)      In Figures 3-5 I observe some peculiar behavior of magnitude of the resonances, which I don’t understand quite well. Magnitude of the dipole resonance in Fig 3b (no gain) is about 2.5e-3, in Fig 4b (G=0.02) it rises to ~6e-3, and in Fig 5b (G=0.03) it again drops to ~3e-3 (seemingly, because ω21 is tuned there to the quadrupole resonance). For the quadrupole resonance, its magnitude in Fig 3d (no gain) is about 2e-7, in Fig 4d (G=0.02) it stays about the same 2e-7 (because ω21 is tuned there to the dipole resonance), and in Fig 5d (G=0.03) it raises greatly to ~1.2e-6 (that is the most interesting result presented in the manuscript). From where does such difference in behaviors of magnitudes come? Is it valid to say that optical modes of higher orders are more sensitive to the gain of the surrounding medium? Why? Please, address the abovementioned questions in the text.

It might be interesting to see dependences of magnitudes of |a1| and |a2| on G up to Gth (at the corresponding ω21) as a separate figure. Also, values of |a1| at the frequency of the quadrupole resonance for different G can be useful for further discussion, for the authors often mention that for G=0.03 a2 exceeds a1 at the quadrupole resonance frequency, but nowhere present the value of the latter.

6)      Basically, Fig 6a duplicates results presented in Fig 3b,d, as well as Fig 7a duplicates Fig 5b,d. I would place Fig 6b,7b together as a separate figure and omit a-parts (at the discretion of the authors). Also, Fig 6b,7b are difficult to comprehend. I encourage the authors to think about better ways of representation the idea. Maybe use radiation pattern instead of streamlines to identify the dominant mode (examples of representation modes via radiation patterns can be found in, e.g., 10.1364/JOSAB.21.001328, 10.1103/PhysRevB.99.075425, 10.1021/acs.jpcc.8b03485). Also, it might be useful for demonstration purposes to find a regime, if any, with even higher quadrupole impact. Also, some design remarks: Fig 6a,7a, if left, should be in the same style as other graphs (e.g. no gray background, etc.).

7)      Minor notices:

·         typo in line 106 «(namely a1 and a

·         The writing is overall fine, though somewhere too long sentences make it difficult to follow the idea. I encourage the authors to revisit the text and split some extensively long sentences into 2 or even 3.

Author Response

We can't thank enough Referee 2 for the thorough revision of our paper. We believe it was extremely useful to improve the quality of our work. Taking into account their advice, we revised all of the figures presented; some were changed in order to make them neater, some were merged to make the ideas discussed more coherent, and a new image was added. The sentences were edited and shortened so they would be easier to follow, as was suggested.

1. In section 2.2 we added three more equations to elaborate on the physical meaning of the expansions used Eqs. (11-13). Then, we depict the field inside the metal particle Em as the internal field Ein Eq. (14), and the field in the gain medium Eh as the sum of the incident and scattered fields Einc and Esca Eq. (15). We also specified the fact that Einc was modeled as a plane wave. Further, we explain the meaning of indices (1) and (3) of the vector spherical harmonics. The typo in Eq. (11) was corrected.

2. In section 3.1 we now discuss the relation between the imaginary part of the steady state polarizability becoming negative, and the emergence of an emissive regime. For this reason, and also to keep continuity with the work on reference [34], we kept the former representation of both the time evolution, and the real and imaginary parts.

3.In Fig 1. we changed the style and color of the lines of the real and imaginary parts of the Polarizability to make the one-to-one correspondence the Quasi Static and Mie approximations have in a 1 nm particle much clearer.

4. While Fig. (3) serves the purpose of depicting how when no gain is added, even if we choose the emission center-line ω21 close to each mode's resonance frequency, the quadrupolar term is much smaller than the dipolar one, it is fundamental later in the same section when we compare it to Fig. (4). In Fig. (4) after adding gain to the system, graphs 4(a-d) showcase the resonance ω21 centered close to the dipolar mode, while (e-h) depict a resonance ω21 centered close to the quadrupolar mode. We find that while we expect overall enhancement when gain is added to the system, it does not happen equally to every mode. Here Fig. (3) comes in useful to notice that the effect of adding gain to a system with dipolar resonance frequency is a threefold amplification in the a1 coefficient, while a2 stays almost the same (comparing 3(b,d) with 4(b,d)). Instead, when the resonance frequency is quadrupolar, we notice a sixfold increase in the coefficient a2, while a1 has a small enhancement but stays in the same order of magnitude (comparing 3(b,d) with 4(d,h)). For these reasons, and also, as we later discuss in section 3.2, since we discovered a relation between the real part of a generic Mie coefficient becoming negative and the emergence of emission, we thought it was convenient to keep the former version of Fig. (3). However, we noticed it was necessary to rewrite the section in order to make all of these findings and the relations between the figures much more apparent, so we rephrased our arguments in this section.

5. Regarding the larger amplification the referee notice on the quadrupolar mode (with respect to the dipolar), when the same amount of gain is added to the system, as suggested by the referee, as suggested by the referee we explicitly mentioned this effect on the text and we included some hypotheses, however in order to produce a solid and scientifically sound explanation for this different behavior a larger and more specific characterization would be needed. We believe this to be beyond the scope of the current work and we commit to discuss it properly in a following publication. In order to further exemplify how an emission center-line ω21 can be placed closer to the resonance frequency of the mode one wants to amplify (in this case the resonance frequency of a2) we included the figure suggested by the referee. This figure now shows the dependencies of the square magnitudes of a1 and a2 as a function of G in a range up to Gth (and corresponding to ω21). We thank the referee for this suggestion for it improves a lot the flow of information.

6. We see the referee's point about the mentioned figures; however, we still believe that one important piece of information here is the comparison between the amplitude of the quadrupolar mode with the tail of the dipolar one at the frequency of the former. We humbly ask the referee to consider that, as they themselves mention, an important point here is to show that the quadrupolar mode can be made larger than the dipolar one, for this reason we believe that it is important to show them on the same scale. In order to further enhance this effect we added to the mentioned figures an inset zooming on the quadrupole frequency range.
We appreciate the suggestion of the referee to use radiation pattern to represent the way the field changes as a consequence of the gain injection; however we still believe that the streamline representation is more suitable to manifest the "sculpturing" effect this injection has on the scattered field. We will consider to include the radiation patterns in the forthcoming works. Finally, as suggested by the referee, we updated the mentioned figures in order to match in style the other graphs.

7. The typo was corrected

Reviewer 3 Report

1] Conclusion is missing??

2] Abstract looks like just a review article, remove extensive general information and add numerical values obtained from the results

3] Resolution of Figures is poor as their legends are not clearly visible

4] Highlight the novelty and significant of your work in the last paragraph of introduction 

Author Response

1. A conclusion was added.

2. We thank Reviewer 3 for this observation, the last sentence of the Abstract was in fact a little cryptic and, in its former version, not very clear in presenting the novelty and the value of our work. We deeply rephrased it in order to better spotlight the relevance of our results.

3. We enhanced the resolution of the images as was suggested, and also improved the use of space by making most of the images larger, advancing quality and readability.

4. As suggested by the referee, we highlighted the novelty and significance of our work in the additional paragraph from line 45 to 59 of the introduction.

Round 2

Reviewer 3 Report

It can be accepted now